

# Increased microRNA-30a levels in bronchoalveolar lavage fluid as a diagnostic biomarker for lung cancer

Wen-Jun Zhu[1,2,*], Bo-Jiang Chen[1,*], Ying-Ying Zhu[1], Ling Sun[1,2], Yu-Chen Zhang[1,2], Huan Liu[3] and Feng-Ming Luo[1,2]

[1] Department of Respiratory and Critical Care Medicine, West China Hospital, Sichuan University, Chengdu, China

[2] Laboratory of Pulmonary Immunology and Inflammation, Frontiers Science Center for Disease-related Molecular Network, Sichuan University, Chengdu, Sichuan, China

[3] Department of General Medicine, University-Town Hospital of Chongqing Medical University, Chongqing, China

[*] These authors contributed equally to this work.

## ABSTRACT

**Background**. MicroRNA-30a (miRNA-30a) levels have been shown to increase in the plasma of lung cancer patients. Herein, we evaluated the miRNA-30a levels in the bronchoalveolar lavage fluid (BALF) of lung cancer patients as a potential biomarker for lung cancer diagnosis.

**Methods**. BALF miRNA-30a expression of 174 subjects was quantified using quantitative real-time reverse transcription-polymerase chain reaction and compared between lung cancer patients and control patients with benign lung diseases. Moreover, its diagnostic value was evaluated by performing receiver operating characteristic (ROC) curve analysis.

**Results**. The relative BALF miRNA-30a expression was significantly higher in the lung cancer patients than in the controls ($0.74 \pm 0.55$ versus $0.07 \pm 0.48$, respectively, $p < 0.001$) as well as in lung cancer patients with stage I–IIA disease than in those with stage IIB–IV disease ($0.98 \pm 0.64$ versus $0.66 \pm 0.54$, respectively, $p < 0.05$). Additionally, miRNA-30a distinguished benign lung diseases from lung cancers, with an area under the ROC curve (AUC) of 0.822. ROC analysis also revealed an AUC of 0.875 for the Youden index-based optimal cut-off points for stage I–IIA adenocarcinoma. Thus, increased miRNA-30a levels in BALF may be a useful biomarker for non-small-cell lung cancer diagnosis.

## INTRODUCTION

Lung cancer is the leading cause of cancer-related mortality worldwide (*Torre, Siegel & Jemal, 2016*). The World Health Organization estimates that the trends of lung cancer deaths will continue to rise, largely due to an increase in global tobacco use, especially in Asia (*Dela Cruz, Tanoue & Matthay, 2011*). In addition, lung cancer is usually diagnosed at an advanced stage because of the lack of clinical symptoms (*Tissot et al., 2015*), leading

Corresponding author
Feng-Ming Luo,
fengmingluo@outlook.com

to a poor prognosis (*Alberg et al., 2013*). Given its low rate of early diagnosis, lung cancer not only seriously affects the survival and prognosis of patients but also imposes a heavy social and economic burden. Therefore, it is essential to explore new biomarkers for early detection of lung cancer.

Biopsy under flexible bronchoscopy (FB) is now established as an essential diagnostic tool for patients with abnormal imaging or sputum findings, according to the routine lung cancer screening process (*Du Rand et al., 2013*). The diagnostic yield from FB ranges from 48% to 93%, depending on whether the tumor is visible within the bronchial tree (*Rivera & Mehta, 2007*). However, biopsy with FB is limited when the lesions are with obviously exposed blood vessels and when patients with clinical risk factors for bleeding (such as anticoagulation therapy, renal dysfunction) (*Du Rand et al., 2013*). Bronchial washings with FB is an alternation when biopsy is limited, which increases the diagnostic yield of FB in patients with lung cancer (*McLean et al., 1998*). Therefore, alternative diagnostic process should be taken into account for individuals undergoing FB but with invisible tumor in the bronchial tree or risk factors of biopsy.

MicroRNAs (miRNAs) are a class of 22-nucleotide long, non-coding, single-stranded RNAs (*Yi et al., 2003*). There is increasing evidence that miRNA expression is associated with various human cancers and may function as tumor suppressors or oncogenes (*Bridge et al., 2012*). MiRNAs play an important role in the cancer cells' development (*Bussing, Slack & Grosshans, 2008*; *Johnson et al., 2007*), invasion, metastasis (*Yao et al., 2010*), proliferation, differentiation, apoptosis, and neoplastic transformation (*Bartel, 2004*; *Boehm & Slack, 2005*), indicating that miRNAs may serve as a new source of biomarkers for the diagnosis of tumors (*Chen et al., 2008*; *Esquela-Kerscher & Slack, 2006*; *Fredsøe et al., 2018*; *Kim et al., 2015*; *Mitchell et al., 2008*).

MiRNA-30a is a member of the miRNA-30 family, which contains five distinct premature miRNAs sequences (miR-30a, -30b, -30c, -30d and -30e) (*Bridge et al., 2012*; *Mao et al., 2018*). Increasing evidence suggested that circulating miRNA-30a has been correlated with several types of human cancers (*Fredsøe et al., 2018*; *Zeng et al., 2013*; *Zhou et al., 2015*). miRNA-30a has been found to be related to the transformation, prognosis, and diagnosis of lung cancer (*Kumarswamy et al., 2012*; *Tang et al., 2015*; *Wen et al., 2015a*), which indicates miRNA-30a may act as a biomarker for lung cancer detection. Our previous study demonstrated that plasma miRNA-30a levels increased in non-small-cell lung cancer (NSCLC) with a relatively high value in the diagnosis of NSCLC (*Sun et al., 2016*). However, the expression level of miRNA-30a levels in bronchoalveolar lavage fluid (BALF) and diagnostic value of BALF miRNA-30a for those individuals suspected for lung cancer with indications of flexible bronchoscopy but with relatively high risk to perform biopsy still remains unknown.

In this study, we investigated the relative expression of miRNA-30a in BALF and explored its diagnostic value for lung cancer, especially at an early stage.

## MATERIAL & METHODS

### Study setting and subjects

The protocol for this prospective study was approved by the Medical Ethics Committee of the West China Hospital, Sichuan University (approval number 2018-83). Samples were collected and processed in a double-blind manner. All patients were grouped based on the pathological diagnosis after miRNA-30a levels were measured, and all other data were collected. By the end of this study, 99 patients with histologically proven primary lung cancer, 55 patients with benign lung diseases 20 healthy controls were recruited between December 2015 and February 2017 at our institution. All the patients have not undergone any types of therapy before sample collection.

The clinical data were acquired for each subject from electronic medical records and included information regarding a complete history and physical examination, serological examination, diagnostic imaging scans, and pathology reports. Epidemiological data were also collected using a self-reported questionnaire, which included information such as demographics, performance status, occupational history, and previous medical history.

### Sample collection

Written informed consent was obtained from all patients prior to bronchoscopy. Plasma and BALF collection, miRNA extraction and analysis were performed based on the same SOP (Standard Operating Procedure) for each patient. Briefly, fasting peripheral venous blood (5 ml) and BALF (8 ml) were collected within an hour before and during bronchoscopy process respectively. For patient with lung cancer, BALF specimens were collected from both the sides of the chest lung, including where the tumor was located and the contralateral side (*Kim et al., 2015*). The blood sample and BALF specimens were centrifuged immediately after collection, at 4 °C 3,000 rpm 10 min and 4 °C 2,000 rpm 10 min respectively. The supernatants were stored at 4 °C temporally and were transferred to −80 °C within one hour. RNA sample extraction, Reverse transcription (RT) Quantitative real-time polymerase chain reaction (qPCR) and ELISA (enzyme-linked immunosorbent assay) procedure were performed within one week after BALF and plasma sample collection.

RNA quality was measured by RNA concentration, purity and integrity. RNA concentration and purity was detected by measuring the absorbance at 260 nm/280 nm (A260/A280) and 260 nm/230 nm (A260/A230) in a Nanodrop 1,000 spectrophotometer (Thermo Fisher Scientific Inc., DE, USA). RNA integrity was measured by 1%AGE (Agarose Gel Electrophoresis). With the quality of high purification(A260/A280 was in 1.8−2.1, A260/A230 > 1.8) and integrate bands, the RNA extracted is suitable for molecular biology operation.

### Measurement of miRNA-30a levels

Total RNA was isolated from BALF specimens using the TRIzol-based method (Life Technologies, Carlsbad, CA, USA), and the RNA pellet was dissolved in 30 μL of RNAase-free water as previously reported (*Sun et al., 2016*). The purity and concentration of RNA were evaluated using a ScanDrop 100 spectrophotometer (Analytik Jena AG, Jena,

Germany). Reverse transcription (RT) was performed using a RevertAid RT kit (Thermo Fisher Scientific, Waltham, MA, USA) according to the manufacturer's instructions.

Quantitative real-time polymerase chain reaction (RT-qPCR) assays were performed as previously reported (*Sun et al., 2016*) in a 20 μL volume using a CFX96 real-time PCR system (Bio-Rad, Hercules, CA, USA). The reactions were incubated at 95 °C for 3 min, followed by 40 cycles of 10 s at 95 °C, 15 s at 59 °C, and 10 s at 72 °C. The relative expression levels of miRNA-30a were calculated as expression value of miRNA-30a to the expression value of U6 via the comparative $2^{-\Delta\Delta CT}$ method (*Fredsøe et al., 2018*). Finally, the results were processed and analyzed using the Bio-Rad CFX Manager.

## Statistical analyses

All statistical analyses were performed using SPSS version 21 (SPSS Inc., Chicago, IL, USA). The association of categorical variables was assessed using the chi-squared test. A *t*-test or analysis of variance was used to explore the distribution of continuous variables among the study groups, as appropriate. Receiver operating characteristic (ROC) curve analysis was applied to investigate the potential of miRNA-30a and other tumour markers to distinguish between the benign and malignant groups. The optimal Youden index was used as the cut-off point for lung cancer diagnosis (*Berezikov et al., 2006*) based on the sensitivity and specificity. Values of $p < 0.05$ were considered to be statistically significant.

# RESULTS

## Clinical characteristics of study subjects

BALF samples were collected from 99 patients with histologically proven lung cancer, 55 patients with benign lung diseases and 20 healthy control patients. There were 62 males and 37 females in the lung cancer group, and the mean age was $57.97 \pm 10.33$ years. The pathological types included 60 adenocarcinomas, 22 squamous cell carcinomas, 15 small-cell lung cancers, and two large-cell lung cancers. According to the 8th Lung Cancer TNM Classification and Staging System, 19, 8, 31, and 26 patients with stages I, II, III, and IV lung cancer, respectively, were identified based on the postoperative pathology. Among patients with benign lung diseases, there were 39 males and 16 females with a mean age of $54.53 \pm 10.62$ years, including 47 pathology-verified cases of pneumonia, two cases of inflammatory pseudotumor, and six cases with benign lung nodules. There were 12 males and 8 females in the healthy control group, and the mean age was $53.05 \pm 13.20$. Patients with benign lung diseases as well as patients in the healthy control group had not developed any types of cancer during the whole follow-up period. There were no statistically significant differences in the clinical characteristics among three groups (Table 1).

## BALF miRNA-30a levels in three groups

MiRNA-30a was significantly upregulated in BALF specimens collected from the lesion side of lung cancer patients compared with patients with lung benign lesions and healthy controls (all, $p < 0.001$) (Fig. 1).

**Table 1 Demographic characteristics of study subjects.**

|  | Lung cancer cases ($n = 99$) | Benign control cases ($n = 55$) | Healthy control ($n = 20$) | *p*-value |
|---|---|---|---|---|
| Sex |  |  |  |  |
| male: female | 62:37 | 39:16 | 12:8 | 0.52 |
| Median age(y) | $57.97 \pm 10.33$ | $54.53 \pm 10.62$ | $53.05 \pm 13.20$ | 0.061 |
| BMI, kg m$^{-2}$ | $22.77 \pm 2.80$ | $23.11 \pm 3.97$ | $22.55 \pm 1.94$ | 0.728 |
| Smoking status |  |  |  |  |
| Smoker | 56 (57%) | 26 (47%) | 9 (45%) | 0.426 |
| Never smoked | 43 (43%) | 29 (53%) | 11 (55%) |  |

**Notes.**

Data are presented as $n$, $n(\%)$, or the mean $\pm$ standard deviation.

BMI, body mass index.

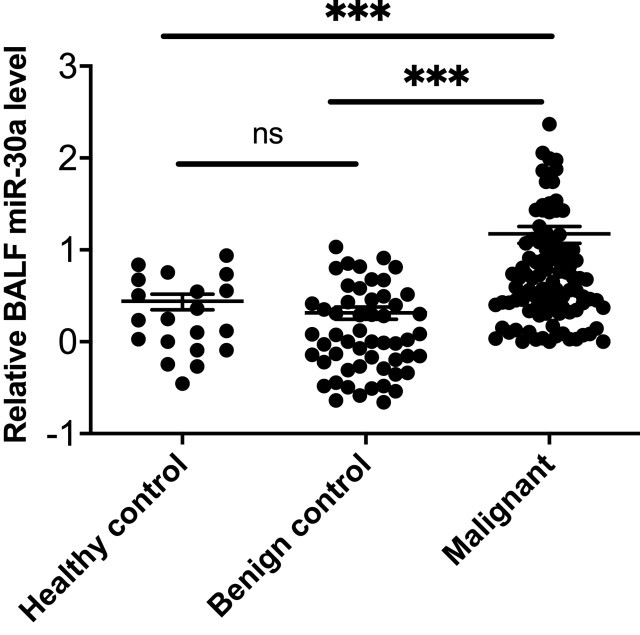

**Figure 1 miRNA-30a levels in BALF samples obtained from lung cancer patients, patients with benign lung diseases and healthy control.** miRNA-30a levels in BALF samples obtained from lung cancer patients, patients with benign lung diseases and healthy control, as determined by quantitative real-time polymerase chain reaction (RT-qPCR) ($n = 55$ in the benign control group, $n = 99$ in the lung cancer group and $n = 20$ in the healthy control). *$p < 0.001$. Data was log transformed to approximate a normal distribution.

## MiRNA-30a levels in BALF specimens obtained from the sides with and without lung cancer lesions

The origin of miRNA-30a in BALF is unknown. It may be secreted by tumour cells, cells adjacent to tumour tissues, immune cells accumulated around the tumour, or even cells circulating in the bloodstream. It is reasonable to hypothesize that the level of miRNA-30a in BALF specimens obtained from the side with the lesion would be higher than in that

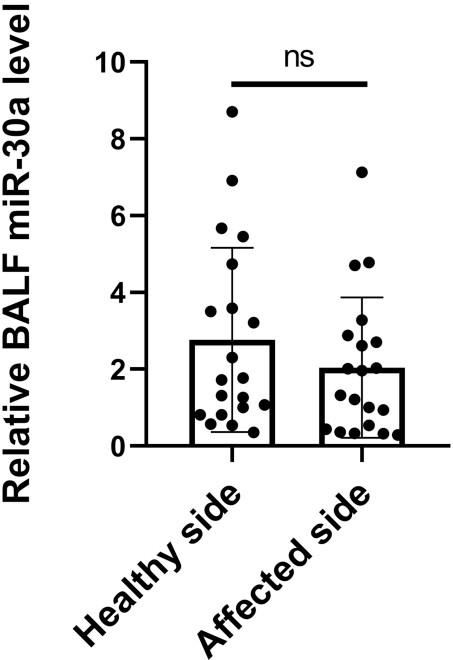

**Figure 2** **miRNA-30a levels in BALF specimens obtained from the sides with and without lung cancer lesions.** ($n = 20$, $p > 0.05$).

obtained from the side without the lesion. Therefore, BALF specimens were collected from lung cancer patients via flexible bronchoscopy from both the sides, with and without the lesions, simultaneously. No statistically significant difference was found in the BALF miRNA-30a levels between the two sides ($0.23 \pm 0.65$, $p = 0.06$) (Fig. 2). Therefore, we subsequently analyzed the level of miRNA-30a in the BALF specimens obtained from the side with the lung cancer lesion.

## Correlation between BALF miRNA-30a levels and clinicopathological features of lung cancer

To explore the possible relationships between BALF miRNA-30a levels and the clinicopathological features of lung cancer, we analyzed the data based on the following factors: age, sex, smoking status, histological type, and TNM stage. No statistically significant differences were found between miRNA-30a levels and these clinicopathological features, apart from the clinical TNM stage (Table 2 and Fig. 3).

## Diagnostic value of miRNA-30a levels in lung cancer patients

Since the level of miRNA-30a in BALF specimens was significantly higher among lung cancer patients compared with that of the controls, miRNA-30a might serve as a biomarker for lung cancer diagnosis. ROC curve analysis was performed to evaluate the diagnostic value of miRNA-30a levels. The area under the curve (AUC) for miRNA-30a was 0.822 (95% CI [0.752–0.892], $p < 0.001$) (Fig. 4).

**Table 2** Relationship between miRNA-30a levels and clinicopathological features among patients with lung cancer.

| Clinicopathological features | Subgroup | Subjects | miRNA-30a relative expression | |
|---|---|---|---|---|
| | | | Mean ± SD | p-value |
| Tissue type | SCLC | 15 | 0.72 ± 0.58 | |
| | NSCLC | 84 | 0.74 ± 0.55 | 0.911 |
| Tissue type subgroup | SCC | 22 | 0.71 ± 0.44 | |
| | ADC | 60 | 0.76 ± 0.59 | |
| | SCLC | 15 | 0.72 ± 0.58 | 0.924 |
| Age, years | ≤60 | 51 | 0.69 ± 0.55 | |
| | >60 | 48 | 0.79 ± 0.55 | 0.387 |
| Sex | Male | 62 | 0.75 ± 0.58 | |
| | Female | 37 | 0.71 ± 0.50 | 0.710 |
| Smoking | Yes | 56 | 0.77 ± 0.59 | |
| | No | 43 | 0.69 ± 0.49 | 0.473 |
| Clinical TNM stage | I–IIA | 22 | 0.98 ± 0.64 | |
| | IIB–IV | 62 | 0.66 ± 0.54 | 0.025[*] |

**Notes.**

SD, standard deviation; SCLC, small-cell lung cancer; NSCLC, non-small-cell lung cancer; SCC, squamous cell carcinoma; ADC, adenocarcinoma.

[*]$p < 0.05$.

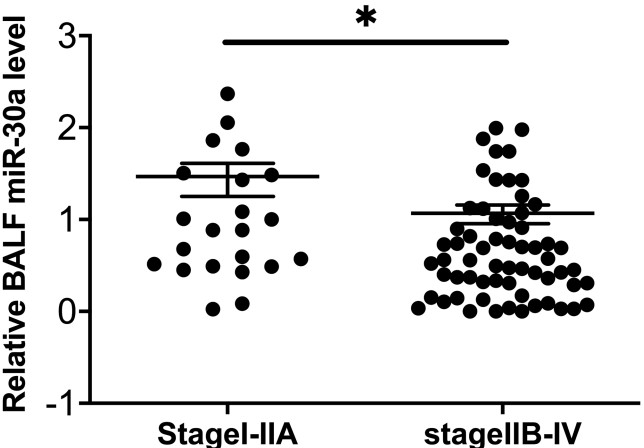

**Figure 3** **Correlation between BALF miRNA-30a levels and TNM stage.** Bronchoalveolar lavage fluid (BALF) miRNA-30a expression was measured by quantitative real-time polymerase chain reaction (RT-qPCR) ($n = 22$ for stages I–IIA and $n = 62$ for stages IIB–IV). *$p < 0.05$. Data was log transformed to approximate a normal distribution.

As shown in Table 3, the best diagnostic cut-off value of relative miRNA-30a levels in BALF was 2.0316 (with a diagnostic sensitivity and specificity of 80.8% and 69.1%, respectively).

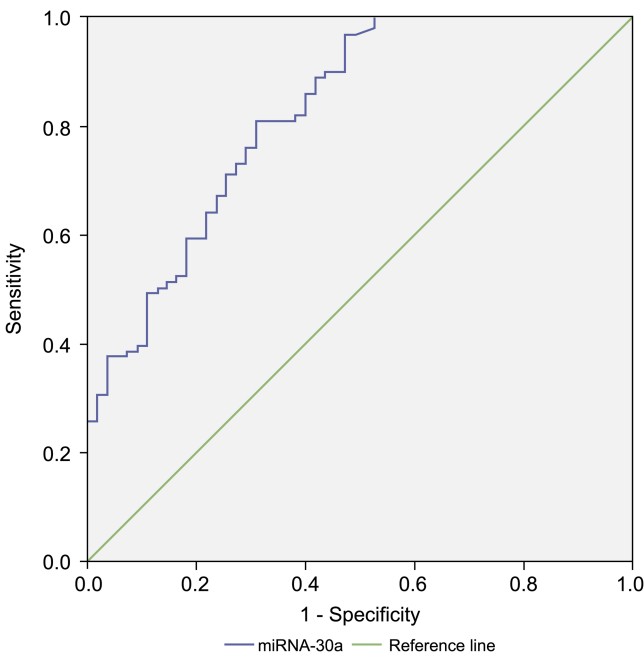

**Figure 4** **Receiver operating characteristic (ROC) curve analysis of miRNA-30a level to distinguish between lung cancer and benign lung diseases.** The area under the curve for miRNA-30a was 0.822, and the Youden index was 0.499.

**Table 3** **Diagnostic value of bronchoalveolar lavage fluid miRNA-30a based on the optimal cut-off point.**

|  | Value | 95% CI |
|---|---|---|
| Cut-off point | 2.0316 | |
| TPR | 80.8% | 0.7140–0.8776 |
| TNR | 69.1% | 0.5503–0.8047 |
| LR+ | 2.61 | 1.7409–3.9260 |
| LR− | 0.27 | 0.1824–0.4228 |
| PPV | 82.4% | 0.7313–0.8917 |
| NPV | 66.6% | 0.5283–0.7825 |

**Notes.**
CI, confidence interval; TPR, true positive rate (sensitivity); TNR, true negative rate (specificity); LR+, positive likelihood ratio; LR−, negative likelihood ratio; PPV, positive predictive value; NPV, negative predictive value.

## Diagnostic value of BALF miRNA-30a level for stages I–IIA lung cancer and benign lung disease

Compared with levels among patients with benign lung diseases, miRNA-30a was significantly upregulated in the BALF of patients with stages I–IIA lung cancer (0.98 ± 0.64 versus 0.07 ± 0.48, respectively, $p < 0.001$) (Fig. 5), with an AUC of 0.882 (95% CI [0.806–0.958], $p < 0.001$) (Fig. 6). The best diagnostic cut-off value of relative miRNA-30a levels in BALF was 2.6508 (with a diagnostic sensitivity and specificity of 90.9% and 74.5%,

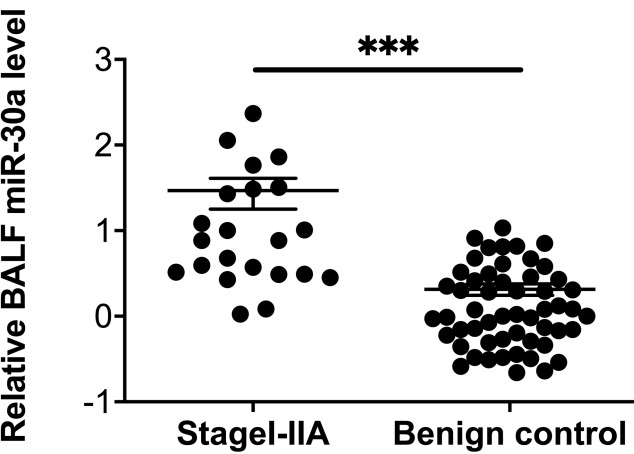

**Figure 5  miRNA-30a levels in stages I–IIA lung cancer patients and benign controls.** miRNA-30a levels in stages I–IIA lung cancer patients ($n = 22$) and benign controls ($n = 55$), as determined by quantitative real-time polymerase chain reaction (RT-qPCR). *$p < 0.001$. Data was log transformed to approximate a normal distribution.

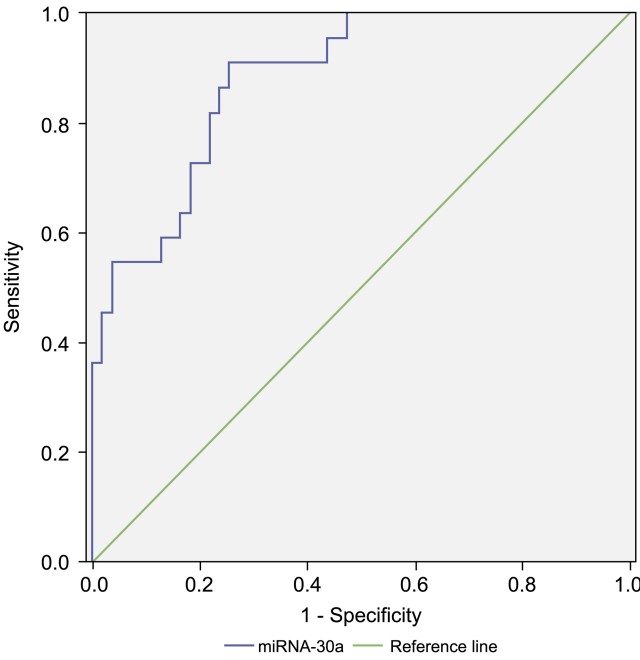

**Figure 6  Receiver operating characteristic (ROC) curve analysis of miRNA-30a level to distinguish between early lung cancer and benign lung diseases.** The area under the curve for miRNA-30a was 0.882 (95% confidence interval: 0.806–0.958, $p < 0.001$), and the Youden index was 0.654.

respectively) (Table 4). This indicated that the upregulated BALF miRNA-30a expression level could be used as an early diagnostic marker for lung cancer.

**Table 4    Diagnostic value of bronchoalveolar lavage fluid miRNA-30a in early (stages I–IIA) lung cancer.**

|  | Value | 95% CI |
|---|---|---|
| Cut-off point | 2.6508 |  |
| TPR | 90.9% | 0.69–0.98 |
| TNR | 74.5% | 0.60–0.845 |
| LR+ | 3.57 | 2.23–5.72 |
| LR− | 0.12 | 0.03–0.46 |
| PPV | 58.8% | 0.41–0.75 |
| NPV | 95.3% | 0.83–0.99 |

**Notes.**
CI, confidence interval; TPR, true positive rate (sensitivity); TNR, true negative rate (specificity); LR+, positive likelihood ratio; LR−, negative likelihood ratio; PPV, positive predictive value; NPV, negative predictive value.

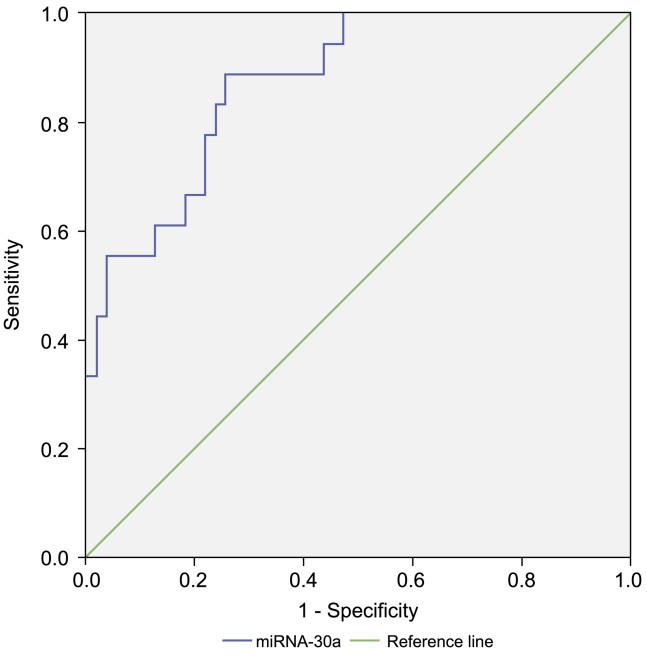

**Figure 7    Receiver operating characteristic (ROC) curve analysis of miRNA-30a level to distinguish between stages I–IIA adenocarcinoma and benign lung diseases.** The area under the curve for miRNA-30a was 0.875 (95% confidence interval: 0.790–0.960, $p < 0.001$).

## Diagnostic value of miRNA-30a expression in BALF among different pathologic types and TNM stages of lung cancer and benign lung disease

There were statistically significant differences in BALF miRNA-30a levels among different pathologic types and TNM stages of lung cancer and benign lung disease ($p < 0.001$). ROC analysis revealed an AUC of 0.875 for the Youden index-based optimal cut-off points for stages I–IIA adenocarcinoma (Fig. 7). Thus, increased levels of miRNA-30a in BALF may serve as a diagnostic biomarker for stage I–IIA adenocarcinoma (Tables 5 and 6).

**Table 5  Diagnostic value of bronchoalveolar lavage fluid miRNA-30a for different pathologic types of lung cancer.**

| Category | Subjects | AUC | *p*-value | 95% CI |
|---|---|---|---|---|
| ADC versus control | 60:55 | 0.821 | <0.001 | 0.75–0.90 |
| SCC versus control | 22:55 | 0.832 | <0.001 | 0.74–0.92 |
| SCLC versus control | 15:55 | 0.817 | <0.001 | 0.71–0.93 |
| NSCLC versus control | 84:55 | 0.823 | <0.001 | 0.75–0.89 |

Notes.
AUC, area under the receiver operating characteristic curve; CI, confidence interval; ADC, adenocarcinoma; SCC, squamous cell carcinoma; SCLC, small-cell lung cancer; NSCLC, non-small-cell lung cancer.
Control group (benign disease, $n = 55$).

**Table 6  Diagnostic value of bronchoalveolar lavage fluid miRNA-30a in different pathologic types and TNM stages of lung cancer.**

| Tissue type | Subjects | AUC | *p*-value | 95% CI |
|---|---|---|---|---|
| ADC, stages I–IIA | 18 | 0.875 | <0.001 | 0.79–0.96 |
| NSCLC, stages I–IIA | 21 | 0.876 | <0.001 | 0.80–0.96 |
| SCC, stages IIB–IV | 14 | 0.822 | <0.001 | 0.72–0.93 |
| SCLC, stages IIB–IV | 10 | 0.773 | <0.05 | 0.63–0.91 |
| ADC, stages IIB–IV | 37 | 0.783 | <0.001 | 0.69–0.87 |
| NSCLC, stages IIB–IV | 52 | 0.796 | <0.001 | 0.71–0.88 |

Notes.
AUC, area under the receiver operating characteristic curve; CI, confidence interval; ADC, adenocarcinoma; NSCLC, non-small-cell lung cancer; SCC, squamous cell carcinoma; SCLC, small-cell lung cancer.

## Combined diagnostic value of BALF miRNA-30a and plasma tumour markers

In addition, we compared the diagnostic value of BALF miRNA-30a with that of traditional plasma tumour markers (carcinoembryonic antigen [CEA] and CYFRA21-1) based on the ability to differentiate lung cancer patients from patients with benign lung disease. We determined the BALF miRNA-30a, plasma CEA, and CYFRA21-1 levels in stages I–IIA NSCLC and control groups and found that the AUC of miRNA-30a was larger than that of CEA and CYFRA21-1. A combination of BALF miRNA-30a levels with plasma CEA and CYFRA21-1 levels could improve the diagnostic value for advanced NSCLC. When plasma CYFRA21-1, CEA, and BALF miRNA-30a levels were combined, the AUC reached the maximum value of 90.3% for stages I–IIA NSCLC and 92.3% for stages IIB–IV NSCLC, which makes this combination preferable for the diagnosis of NSCLC (Table 7 and Fig. 8).

## DISCUSSION

Our previous data showed miRNA-30a was down-regulated and up-regulated in lung cancer tissue (*Sun et al., 2016*) and plasma respectively (*Liang et al., 2019*), which arouse our interest in analyzing miRNA-30a expression in BALF of lung cancer. In this hospital-based case-control study, we found the expression level of miRNA-30a in BALF was significantly up-regulated in lung cancer patients when both compared to those with benign lung disease and healthy control patients (both $p < 0.001$). The expression level

**Table 7   Diagnostic value of bronchoalveolar lavage fluid miRNA-30a in combination with plasma tumor markers.**

| Diagnostic value | Subjects | Subgroup | AUC | *p*-value | 95% CI |
|---|---|---|---|---|---|
| ADC, stages I–IIA | 10 | miR-30a | 0.910 | <0.001 | 0.81–1.00 |
| | | CEA | 0.572 | >0.05 | 0.36–0.78 |
| | | CEA + miR-30a | 0.903 | <0.001 | 0.80–1.00 |
| ADC, stages IIB–IV | 29 | miR-30a | 0.790 | <0.001 | 0.68–0.91 |
| | | CEA | 0.840 | <0.001 | 0.74–0.94 |
| | | CEA + miR-30a | 0.880 | <0.001 | 0.80–0.97 |
| NSCLC, stages I–IIA | 12 | miR-30a | 0.882 | <0.001 | 0.78–0.99 |
| | | Cyfra21-1 | 0.438 | >0.05 | 0.22–0.65 |
| | | Cyfra 21-1 + miR-30a | 0.885 | <0.001 | 0.78–0.99 |
| NSCLC, stages IIB–IV | 34 | miR-30a | 0.796 | <0.001 | 0.69–0.91 |
| | | Cyfra21-1 | 0.813 | <0.001 | 0.71–0.92 |
| | | Cyfra21-1+miR-30a | 0.868 | <0.001 | 0.78–0.96 |

**Notes.**
AUC, area under the receiver operating characteristic curve; CI, confidence interval; CEA, carcinoembryonic antigen; ADC, adenocarcinoma; NSCLC, non-small-cell lung cancer.

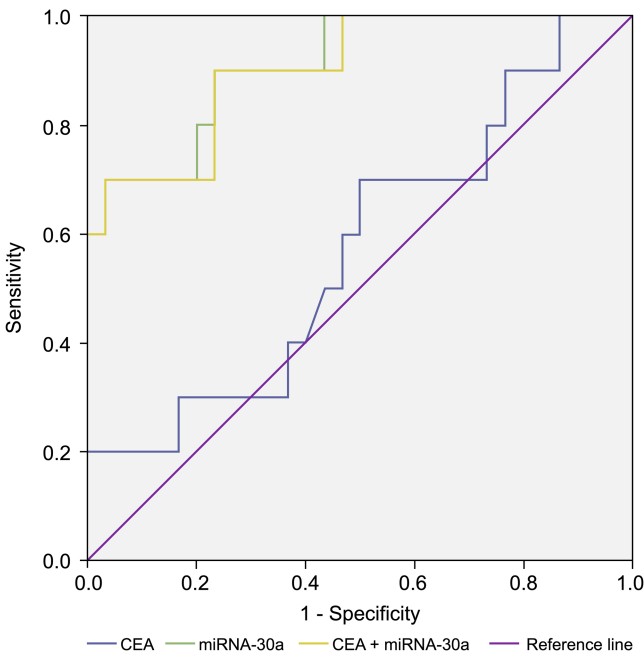

**Figure 8   Receiver operating characteristic (ROC) curve analysis.** Receiver operating characteristic (ROC) curve analysis of miRNA-30a level, CEA level, and a combination of miRNA-30a and carcinoembryonic antigen (CEA) levels to distinguish between stages I–IIA adenocarcinoma and benign lung diseases. The area under the curve for miRNA-30a was 0.910 (95% CI [0.808–1.000], *p* < 0.001).

of miRNA-30a in BALF was significantly higher in individuals with early stage (stage I–IIA) of lung cancer when compared to those with stage IIB–IV ($p < 0.05$). The ROC curve analysis revealed that the BALF miRNA-30a level could well distinguish lung benign disease and lung cancer as well as early stage of lung cancer with the relative high AUC of 0.822 and 0.882 respectively. The diagnostic value was even higher when combined BALF miRNA-30a, CEA and CYFRA21-1. All those findings indicated that BALF miRNA-30a relative expression level could be used as a biomarker in early detection of lung cancer for individuals undergoing FB but with invisible tumor in the bronchial tree or risk factors of biopsy.

Abundant circulation miRNAs have been demonstrated to play a vital role in early cancer detection (*Kim et al., 2015*; *Mitchell et al., 2008*; *Tsujiura et al., 2010*). The circulating cell-free miRNAs have shown great promise as a new class of cancer biomarkers, owing to their surprisingly high stability in body fluids, association with disease states, and ease of measurement (*Heinzelmann et al., 2011*; *Zheng et al., 2011*). MiRNA-30a, as a member of the miRNA-30 family, has been correlated with several types of human cancers (*Saleh et al., 2019*; *Sun et al., 2019*; *Wen et al., 2015b*). The expression level of miRNA-30a was down-regulated in breast cancer (*Zeng et al., 2013*) but up-regulated in prostate cancer and ovarian serous adenocarcinoma (*Fredsøe et al., 2018*; *Zhou et al., 2015*). Our present study revealed that miRNA-30a in BALF was correlated with different stages of lung cancer. The opposite activity of miR-30a in various cancer type and different stages in one cancer type indicated that the biological function of miR-30a might be complicated and associated with a specific cancer type and tumor microenvironment, which needs to be further explored.

There are several potential mechanism of miRNA-30a in the development, invasion and metastasis of lung cancer. First, miRNA-30a inhibits the epithelial-to-mesenchymal transition by targeting SNAI1, and thus, its downregulation in lung cancer results in cancer invasion and metastasis (*Kumarswamy et al., 2012*). Second, *Guan, Rao & Chen (2017)* have found that miR-30a suppresses lung cancer progression by targeting SIRT1. In addition, expression of miR-30a inhibitor the growth of lung cancer cells by targeting MEF2D (*Luan, Wang & Liu, 2018*). Further studies are necessary to identify the target genes of miR-30a and elucidate the underlying mechanism that regulates the biogenesis of miR-30a.

The expression level of miRNA-30a in body fluids, including BALF and the plasma (*Liang et al., 2019*) of lung cancer patients, were significantly upregulated, while the opposite trend was observed in lung cancer tissue (*Boeri et al., 2011*; *Kumarswamy et al., 2012*; *Tang et al., 2015*). There are several possible mechanisms. microRNAs are transported into the extracellular fluid environment from cells by exosomes and microvesicles (*Etheridge et al., 2011*). *Pritchard et al. (2012)* have presented evidence showing that blood cells are a major contributor to circulating miRNAs. The extracellular circulation of miRNAs in lung cancer patients may be due to a response of blood cells to the tumour state (*Patnaik et al., 2012*) and shifting to various body fluids. However, the exact mechanism still needs to be investigated. However, the exact biological basis of this discrepancy between miRNA-30a expression changes in cancer tissues and in circulating body fluids in lung cancer still remains unclear and needs to be investigated in the future research.

The present study does have some limitations. First, as a hospital-based case-control study, the relatively small sample size of each group was unavoidable. Future studies with larger cohorts are warranted to validate these findings. Second, there was no significance in BALF miRNA-30a expression levels among the adenocarcinoma, squamous cell carcinoma and small cell lung cancer, and it may be related to the relative small sample size of small cell lung cancer. Third the underlying molecular mechanism that accounts for the contradiction between miRNA-30a expression in cancer tissues and in circulating body fluids of lung cancer needs further investigation.

## CONCLUSIONS

In conclusion, BALF miRNA-30a was significantly up-regulated in lung cancer patients and correlated with early stage lung cancer. These present findings indicated that BALF miRNA-30a relative expression level could be used as a biomarker in early detection of lung cancer for individuals undergoing FB but with invisible tumor in the bronchial tree or risk factors of biopsy.

## ACKNOWLEDGEMENTS

The authors thank Prof Huajing Wan for advising on the article revision. The authors thank the patients who took part in the study and the staff of the Department of Pulmonary and Critical Care Medicine, West China Hospital, Sichuan University for their help with data collection.

### Funding

This work was supported by the National Nature Science Foundation of China grant (NSFC No. 81770072) and the project of Science & Technology Department of Sichuan Province, China (no. 2016JY0043). The funders had no role in study design, data collection and analysis, decision to publish, or preparation of the manuscript.

### Grant Disclosures

The following grant information was disclosed by the authors:
National Nature Science Foundation of China: 81770072.
Science & Technology Department of Sichuan Province, China: 2016JY0043.

### Competing Interests

The authors declare there are no competing interests.

### Author Contributions

- Wen-Jun Zhu conceived and designed the experiments, performed the experiments, analyzed the data, prepared figures and/or tables, and approved the final draft.
- Bo-Jiang Chen conceived and designed the experiments, analyzed the data, prepared figures and/or tables, authored or reviewed drafts of the paper, and approved the final draft.

- Ying-Ying Zhu performed the experiments, analyzed the data, authored or reviewed drafts of the paper, and approved the final draft.
- Ling Sun performed the experiments, analyzed the data, prepared figures and/or tables, and approved the final draft.
- Yu-Chen Zhang analyzed the data, prepared figures and/or tables, and approved the final draft.
- Huan Liu performed the experiments, prepared figures and/or tables, and approved the final draft.
- Feng-Ming Luo conceived and designed the experiments, authored or reviewed drafts of the paper, and approved the final draft.

## Human Ethics

The following information was supplied relating to ethical approvals (i.e., approving body and any reference numbers):

The Medical Ethics Committee of the West China Hospital, Sichuan University approved this research (2018-83).

## Data Availability

The raw measurements are available in the Supplementary File.

## Supplemental Information

Supplemental information for this article can be found online at http://dx.doi.org/10.7717/peerj.11528#supplemental-information.

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
