# Peer review of "Increased microRNA-30a levels in bronchoalveolar lavage fluid as a diagnostic biomarker for lung cancer"

_PeerJ, doi:10.7717/peerj.11528_

## Round 0.1 · original submission · Major Revisions

Please address all concerns and comments from the reviewers.

·

Basic reporting

This article is clearly written and has reported most of the important aspects. It has used most of the references to explain the aim of the study. I have no issues with the figures and tables they used. The study is hypothesis directed.
However, I have few concerns about the reporting that may be required to be included in the paper,
1. The author should report whether the patients have undergone any types of therapy before the sample collection.
2. Also, it is important to report if the patients with benign lung diseases had any other types of cancer present or if those control cases had developed lung cancer later.

Experimental design

Experimental designs are perfect, and the author has described all the experiments properly to direct this study in a specific direction. I feel the following,
1. It will be informative if the author could show whether the miRNA collected at different time points for each patient and whether the plasma and BALF collection performed at the same time.
3. Change in RNA concentration could occur between the time of collection and analysis. It can be included in the article.
4. Comparing the lung tissues from lung cancer patients and benign lung diseases patients to find out if the tissue could show different miRNA level.
5. Healthy controls can be included since lung diseases could alter RNA level.

Validity of the findings

Although there are several other studies that have been performed but I think this study is unique. The data of the article are statistically sound. Comparing different compartments to detect the status of lung cancer can be crucial. I think they have concluded the study nicely.

Additional comments

The study is well planned and designed. It opened up new areas of research to carry out regarding biomarker detection. This study can be persuaded in future with more detailed experiments in preclinical model.

Reviewer 2 ·

Basic reporting

It is quite important to find a dependable and non-invasive biomarker for lung cancer detection and is therefore an important area to address.

Experimental design

While the goal of this study was understandable, there appears to be significant contradiction of the findings of this study with current literature. Therefore, this study has to be further substantiated to be more meaningful.

Validity of the findings

Some of the data could benefit from better representation

Additional comments

Both the discussion and conclusion of the article are very short. Comprehensive introduction about miRNA-30a is required; kindly add information from available databases worldwide. Most of the information on miRNA-30a is different from the authors’ findings. Kindly explain further.

Line 47- 48 – Rewrite with purpose of Flexible bronchoscopy and other methods of detection on lung cancer

Line 69 – 70 – It appears that other publications show that miRNA-30a is decreased in lung cancer, which is contradictory to your findings. Kindly explain.

Is Fig 2, are the error bars SD or SEM, both groups appear to have very large deviation.

Is there a figure for lines 152 and 153?

In table 2, what is the unit of the mean values?

Lines 169 and 170 - As shown in Table 3, the best diagnostic cut-off value of miRNA-30a levels in BALF was 2.0316 (with a diagnostic sensitivity and specificity of 80.8% and 69.1%, respectively) – Where did authors obtain the value of 2.0316 from (graph) and what is the unit?

---

## Round 0.2 · accepted · Accept

The authors have addressed all reviewer comments.

·

Basic reporting

No comment

Experimental design

No comment

Validity of the findings

No comment

Additional comments

I think this study is very detailed and hypothesis driven.

Reviewer 2 ·

Basic reporting

The authors have addressed my concerns

Experimental design

Issues have been addressed

Validity of the findings

I believe the explanation for the contraindication of their findings to existing literature can be accepted for now and hope they publish the exosome findings in the near future o substantiate this

Additional comments

All concerns have been addressed